# Heavy Metal Contamination in Yogurt from Lebanon: Evaluating Lead (Pb) and Cadmium (Cd) Concentrations Across Multiple Regions

**DOI:** 10.3390/toxics13060499

**Published:** 2025-06-13

**Authors:** Sandra Sarkis, Rayan Kashmar, Nikolaos Tzenios, Maha Hoteit, Tony Tannous, Joseph Matta

**Affiliations:** 1Department of Nutrition, Faculty of Pharmacy, Ecole Doctorale Sciences et Santé (EDSS), Medical Sciences Campus, Saint Joseph University of Beirut (USJ), Beirut 1100, Lebanon; 2GEPEA, Energy Systems and Environment Department, IMT Atlantique, Rue Alfred Kastler, 44307 Nantes Cedex 3, France; 3Doctoral School of Science and Technology, Research and Analysis Platform in Environmental Sciences (EDST-PRASE), Beirut 1100, Lebanon; 4Faculty of Public Health, Charisma University, London EC1V 7QE, UK; 5PHENOL Research Group (Public Health Nutrition Program-Lebanon), Faculty of Public Health, Lebanese University, Beirut 1100, Lebanon; m.hoteit@ul.edu.lb; 6Institut National de Santé Publique, d’Epidémiologie Clinique et de Toxicologie-Liban (INSPECT-LB), Beirut 1100, Lebanon; 7Department of Primary Care and Population Health, University of Nicosia Medical School, Nicosia P.O. Box 24005, Cyprus; 8Faculty of Arts and Sciences, University of Balamand, Tripoli P.O. Box 100, Lebanon; tony.tannous@balamand.edu.lb; 9Department of Nutrition, Faculty of Pharmacy, Saint Joseph University of Beirut, Beirut 1100, Lebanon; joseph.matta@usj.edu.lb; 10Industrial Research Institute, Lebanese University Campus, Hadat Baabda, Beirut 1100, Lebanon

**Keywords:** heavy metals, yogurt, Lebanon, public health, environmental pollution

## Abstract

The toxicity of metals such as lead (Pb) and cadmium (Cd) makes heavy metal contamination in food products a major health concern. The aim of this study is to measure the levels of Pb and Cd in yogurt samples that were collected from 11 distinct Lebanese locations and determine their compliance with the legal limits set by the European Commission (0.02 mg/kg for Pb and 0.005 mg/kg for Cd). A total of 165 yogurt samples were examined using atomic absorption spectrophotometry (AAS). Interestingly, the findings showed that 10.9% of samples had Pb concentrations over the allowable limit; the highest amounts were observed in Baalbeck-Hermel (0.118 mg/kg), North Area (0.125 mg/kg), and South Lebanon (0.115 mg/kg). In addition, the highest detected concentrations of Cd were found in North Area (0.094 mg/kg), Baalbeck-Hermel (0.076 mg/kg), and Akkar (0.042 mg/kg), with 14.5% of samples above the limit. The results show regional differences in contamination, which are probably caused by industrial emissions, agricultural activities, and environmental pollution. To better understand the potential health implications, the estimated daily intake (EDI) of Pb and Cd through yogurt consumption was calculated and compared to international health-based guidance values. Although the EDI values suggest a low risk from yogurt alone, the findings highlight the importance of cumulative exposure and emphasize the necessity of regular monitoring and stricter implementation of food safety laws to decrease exposure to heavy metals through dairy intake.

## 1. Introduction

Exposure to heavy metals is one of the biggest risks to the biological system and, in particular, human health [1]. Heavy metals are persistent environmental contaminants that pose considerable health hazards upon accumulation in dietary items. Among these potentially toxic elements (PTEs), Pb and Cd are particularly concerning due to their toxicity and ability to bioaccumulate in the human body [2].

Their selection in this study is based on previous environmental assessments in Lebanon, which have reported alarming levels of these PTEs in soil, water, and vegetation due to industrial emissions, excessive use of agriculture fertilizers and pesticides, uncontrolled dumping of solid waste, and the use of contaminated water or feed in livestock farming [3,4,5]. In a study done by Moustafa et al. [3], Cd and Pb were found to be the most prevalent pollutants in soil and groundwater assessed in the area of Akkar northern Lebanon. In another study by Borjac et al. [4] in Deir Kanoun Southern Lebanon, Cd and Pb concentrations at several locations exceeded international regulatory thresholds in soil samples collected from the dump and along the canal in the area. Similarly, in the Central Bekaa Valley, ecological risk assessments based on soil analyses classified cadmium (Cd) and lead (Pb) as posing high to moderate environmental threats, while chromium (Cr), nickel (Ni), copper (Cu), and zinc (Zn) were associated with low risk levels [5].

The impact of heavy metals from food origin on human health was explained and summarized by a review done by Peralta-Videa et al. [6]. In a study done by Unicef [7], Pb is reported as a neurotoxin that affects cognitive development, while Cd can harm the kidneys, liver, and bones, and affect the female reproductive system as well [6]. According to the International Agency for Research on Cancer (IARC), Cd is a Group 1 carcinogen [8]. These potentially toxic elements can bioaccumulate as a result of prolonged food exposure, raising the possibility of chronic toxicity. Industrial emissions, contaminated water, soil, and the use of agricultural inputs such as fertilizers and pesticides are some of the ways that heavy metal contamination can occur. In addition to these findings, global health authorities such as the World Health Organization (WHO) have classified Pb and Cd as major public health threats due to their carcinogenic, nephrotoxic, neurotoxic, and reproductive effects [9,10]. Furthermore, both elements are included on the United States Environmental Protection Agency’s (EPA) list of priority pollutants, underscoring their global significance as hazardous contaminants [11].

Monitoring the amounts of heavy metals in food products is essential to guarantee public health and consumer safety because of the possible health effects [12]. It has long been known that dairy products are among the most essential and nutrient-dense foods. Several key components present in dairy products, including fatty acids, amino acids, minerals, and vitamins, are critical for a child’s development and immune system [13]. Yogurt is one of the dairy products most widely consumed, and is made by a fermentation process by different bacteria using milk as a raw material. Additionally, yogurt provides the human body with proteins, minerals, and vitamins, and its advantage is that it can be consumed by people who have lactose intolerance [14].

Dairy products, including yogurt, are susceptible to contamination because heavy metals in animals bioaccumulate. While trace elements are often non-toxic to human health, excessive consumption of essential elements can potentially be harmful to human well-being [15]. The concentrations of metals (including heavy metals and trace elements) in dairy products may be derived from the contaminated environment and the packing materials utilized during production. As an example, plants used as animal feed have the ability to absorb heavy metals from soil and groundwater, which can subsequently be transmitted into raw milk [16]. For example, through the fertilizer of contaminated crops, Cd can find its way into animal feed [17]. On the other hand, the main source of Pb is industrial contamination, which contaminates soil, water, and animal feed before joining the feed channel [18]. Research done by Bousbia et al. showed that the incidence and amounts of heavy metals in milk and dairy products can vary from one study to another and from one country to another due to different factors such as the route of exposure, animal diet, lactation, and environmental pollution [19].

Milk, as a raw material for all dairy products, is considered to be a source of potentially toxic element (PTE) pollution, as it can become more contaminated and, therefore, more dangerous when it is transferred through different stages (e.g., from production to fermentation to other products or from production to consumption) [20,21]. It is very important to mention that PTEs are not destroyed throughout the manufacturing processes of dairy products; therefore, they can be transported into products such as yogurt [22]. Among the heavy metals, Pb and Cd are considered two highly toxic PTEs that can be found in air, water, soil, and food products [20,23]. They are classified as dangerous PTEs, and their toxicity is well known as one of the major worldwide pollutants [24]. For this reason, it is critical to ensure that these two potentially toxic elements are present in food items at the lowest possible concentrations to maintain food safety and protect consumers [25]. The Commission Regulation (EU) 2023/915, which changes Regulation (EC) No. 1881/2006, states that raw milk and dairy products can contain up to 0.020 mg/kg and 0.005 mg/kg of lead and Cd, respectively [26].

Different studies reported the quantification of heavy metals, especially Pb and Cd, in milk and milk products worldwide. As an example, Er et al. [27] evaluated the Pb levels in milk and yogurt samples in Turkey. They found that Pb levels were in the range of 15–61 ng/mL and 21–42 ng/g for milk and yogurt, respectively. Additionally, Winiarska-Mieczan et al. [22] determined the Pb and Cd in yogurts and cream cheese samples in Poland. In addition, Zhou et al. [28] found that the levels of Pb and Cd were higher than the acceptable levels in milk samples, and this was attributed to industrial activities that probably led to milk contamination in China. Yi-Jie Dai et al. [29] reported the Pb and Cd contamination of Mish and Karish cheese samples, respectively, in Egypt. Rebellato et al. [30] determined Pb and Cd levels in yogurt samples in Brazil, and they were 21.58 and 4.20 µg/kg, respectively.

This study aims to evaluate the levels of Pb and Cd in Lebanese yogurt samples collected from various regions and assess their compliance with regulatory limits. In addition, it also estimates the dietary intake of Pb and Cd through yogurt consumption in the Lebanese population. The findings of this study provide valuable insights into the potential health risks associated with yogurt consumption in Lebanon. To the best of our knowledge, no previous studies have been conducted to assess the presence of toxic metals, in particular, Pb and Cd, in yogurt samples across different regions in Lebanon, nor have they estimated consumer exposure based on dietary intake data through yogurt consumption.

## 2. Materials and Methods

### 2.1. Lebanon

Lebanon is located on the eastern Mediterranean coast between 33°03′–34°45′ N latitude and 35°05′–36°30′9″ E longitude. From west to east, Lebanon can be geographically divided into four areas: a coastal plain along the west, the Mount Lebanon mountain chain, the fertile Bekaa Plain, and the Anti-Lebanon mountain chain that runs along the eastern border with Syria; the fertile Bekaa Plain lies between the two mountain chains [31]. The area of Lebanon is 10,452 km^2^ and its population is 5,364,482 people (2024 est.) [32]; hence, Lebanon has a high population density of 513 persons/km^2^. In 2017, industry contributed 13.1% to the gross domestic product, and agriculture, which employed more than 39% of the population in 2009 [33], contributed 3% [34]. Both of these sectors are considered to be major contributors to heavy metal contamination in the environment [35].

### 2.2. Sampling

Yogurt samples were collected from 11 different Lebanese regions: Nabatiyeh (Na), South Area (S), Chouf (C), Metn (M), Kesserwan (K), Jbeil (J), Batroun (B), North Area (No), Akkar (A), Bekaa (Be), and Baalbeck-Hermel (BH). A total of 15 samples were obtained from different dairy farms in each region, resulting in 165 samples. All yogurt products were produced using traditional methods, within the farm’s premises without the addition of preservatives, stabilizers, or flavoring agents, and were made exclusively from cow’s milk sourced from the same region of production. These yogurts reflect the naturally processed dairy products commonly consumed in Lebanon, and represent the regional diversity of local milk production practices.

Following collection, the samples were immediately transported under refrigerated and sterile conditions to the laboratory facilities. Upon arrival, the samples were subjected to chemical analysis without delay, in order to preserve their integrity and prevent any potential degradation or alteration in contaminant concentrations. The general nutritional composition of the collected yogurt samples is summarized in Table 1. The geographic distribution of the study areas is illustrated in Figure 1, which also locates Lebanon in relation to the global map for added contextual reference.

### 2.3. Description of Study Areas

To contextualize the variations in heavy metal concentrations observed across the sampled yogurt products, this section provides a brief overview of each of the 11 Lebanese regions from which samples were collected. The descriptions focus on demographics, predominant economic activities, and potential sources of environmental pollution, if any, that could influence heavy metal contamination in dairy products.

Nabatiyeh (Na): Located in southern Lebanon, Nabatiyeh is characterized by a mix of urban and rural areas. The economy is primarily based on agriculture, with tobacco being a significant crop. The region has experienced environmental challenges due to growing urbanization, industrialization, and population growth, which contribute to the presence of contaminants, and in turn degrades freshwater quality and puts the ecosystem at severe risk [36].

South Area (S): This region encompasses parts of southern Lebanon. The primary economic activity relies on agriculture, particularly tobacco and olive cultivation. Environmental pollution arises from agricultural runoff and the dumping or burning of industrial, household, electronic, plastic, and medical waste, which has led to the contamination of soil and water sources [4].

Chouf (C): Situated in the Mount Lebanon Governorate, Chouf is known for its diverse topography and forested areas. The local economy relies on agriculture, including fruit orchards and olive groves. While industrial activity is limited, the use of fertilizers and pesticides in agriculture could be potential sources of environmental contaminants.

Metn (M): Metn is a district in the Mount Lebanon Governorate with a mix of urban centers and rural villages. The area hosts various industries, including manufacturing and construction, which may contribute to environmental pollution [37]. Additionally, the high population density and vehicular traffic can lead to increased air and soil contamination.

Kesserwan(K): Also part of Mount Lebanon, the region’s economy is driven by tourism, agriculture, and light industry. Environmental concerns include pollution from industrial discharges and untreated sewage affecting water quality [37].

Jbeil (O): Also known as Byblos, Jbeil is a coastal district with a rich historical heritage. The economy is driven by tourism, agriculture, fishing, and small-scale industries. Environmental issues include coastal pollution from industrial and domestic waste, as well as challenges related to water resource management [37].

Batroun (B): Batroun is a coastal city in northern Lebanon known for its tourism and agriculture, particularly citrus groves. Environmental concerns include pollution from agricultural runoff and inadequate waste disposal systems [37].

North Area (No): This region includes parts of northern Lebanon with diverse economic activities, including agriculture, industry, and services. Tripoli, the largest and most significant city in the northern area and the second-largest city in Lebanon, contributes to industrial emissions in the region and discharges solid waste into the Mediterranean Sea from its dumpsite. Other sources of pollution in the North Area stem from the mismanagement of sewage, industrial, and hazardous solid waste [37].

Akkar (A): Akkar is Lebanon’s northernmost governorate, characterized by its rural landscape and agricultural economy. Akkar, the second Lebanese agricultural zone after Bekaa plain, is highly contaminated by prohibited agricultural products. Many studies showed severe contamination of Akkar plain groundwater and soil by nitrate, nitrite, and pesticides [38,39,40]. According to El-Fadel et al. [41], the coastal waters in Akkar are contaminated by industrial wastewater discharges.

Bekaa (Be): The Bekaa Valley is Lebanon’s primary agricultural region, producing a significant portion of the country’s crops. However, the area faces environmental issues such as water pollution from agricultural runoff and industrial waste, which can affect soil and water quality. Excess fertilizer input and uncontrolled disposal of refuse may have resulted in soil and ground water contamination with nitrates and heavy metals [42,43].

Baalbeck-Hermel (BH): Located in northeastern Lebanon, the economy is based on agriculture and livestock. Environmental issues include soil and water pollution from agricultural chemicals, as well as contamination resulting from inadequate waste disposal and water scarcity in the region.

### 2.4. Reagents

All reagents used were of analytical grade. Nitric acid (≥65%) and hydrogen peroxide (30%) were obtained from Sigma-Aldrich (Steinheim, Germany). Stock solutions were diluted with ultrapure water to create fresh individual standard solutions. Pb standard solutions were prepared by serial dilution from a 1000 ± 4 mg/L stock solution sourced from Fluka Analytical (Buchs, Switzerland). Cd standards were similarly prepared by diluting a 1000 mg/L Certipur® stock solution obtained from Merck (Darmstadt, Germany). Fresh working standard solutions were prepared prior to each analytical run to ensure accuracy and stability. All glassware and plasticware were thoroughly cleaned before use. Initially, they were soaked overnight in 1% nitric acid, then rinsed three times with ultrapure water and allowed to air dry in a dust-free environment to prevent contamination.

### 2.5. Preparation of Samples

The method was previously developed and validated on chickpeas by Kassouf et al. [44]. Three replicates of 2 g fresh homogenized samples (no freeze-drying was performed) were accurately weighed into a modified polytetrafluoroethylene (PTFE-TFM) microwave bomb vessel. After adding 7 mL of nitric acid and 1 mL of hydrogen peroxide, the sample digestion was performed in a high-performance microwave digestion system (Anton Paar, Multiwave 3000, Graz, Austria) according to the previously described program by Kassouf et al. [44]. After cooling, the final clear digest was transferred to 50 mL polypropylene tubes, diluted to volume with ultrapure water, and filtered through 4 µm PTFE syringe filters (Whatman, Maidstone, UK) to remove particulates before analysis.

### 2.6. Metal Analysis

Quantification of Pb and Cd was performed using graphite furnace atomic absorption spectrometry (GFAAS) with a Shimadzu AA-6800 atomic absorption spectrometer equipped with an ASC-6100 autosampler and WizAArd software for data acquisition. The GFAAS instrument was operated under optimized conditions for each metal, using wavelengths of 283.3 nm for Pb and 228.8 nm for Cd, and a lamp current of 10 mA. External calibration curves were generated using at least five concentration levels for each metal, prepared fresh from stock solutions.

### 2.7. Method Validation

White cabbage certified reference material (BCR^®^ -679, Sigma Aldrich, Geel, Belgium) was used to control the accuracy of sample treatment and analysis of Cd with a recovery in the range of 94–111%. On the other hand, the accuracy for Pb was previously validated by Kassouf et al. on chickpea samples (Kassouf et al., 2013) [44]. Method validation also included assessments of precision, limit of detection (LOD), and limit of quantification (LOQ). LOD and LOQ were estimated from the average signal of blank solutions prepared via the same digestion procedure, with LOD calculated as the average blank signal plus three times the standard deviation and LOQ as the average plus ten times the standard deviation. The resulting values were compatible with expected concentrations in dairy matrices: LODs for Pb and Cd were 0.004 mg/kg and 0.0007 mg/kg, respectively, and LOQs were 0.012 mg/kg and 0.0021 mg/kg, respectively, in agreement with levels typically observed in food samples [44]. Precision, expressed as the relative standard deviation (RSD%) of triplicate spiked sample analyses, was consistently below 15%, confirming the reproducibility.

### 2.8. Estimated Dietary Intake for Pd and Cd

The estimation of daily intake (EDI, μg/kg b.w./day) for Pb and Cd compounds in yogurt samples was conducted using Equation (1):EDI = C × MS/b.w.(1)
where C is the mean concentration of Pd or Cd (mg/kg, ww), MS is the meal size (daily consumption of yogurt, gram per day), and b.w. is the body weight (kg) of the adult consumer. The EDI was calculated for Pd and Cd compounds quantified above the LOQ values in the yogurt samples. According to Nasreddine et al. [45], the mean consumption of yogurt for adults in Lebanon was 68.3 g/day and the average body mass of adult consumers was 73.8 kg [46]. Data on the average consumption of yogurt in children and adolescents have not yet been published in Lebanon; therefore, we will conduct the EDI for Lebanese adult consumers only.

### 2.9. Data Analysis

The obtained data were analyzed using IBM SPSS Statistics V26. Descriptive statistical parameters, including mean and standard deviation, were used to describe and compare the measured levels of Pb and Cd in yogurt samples across different regions. One-sample two-sided *t*-tests were conducted to compare measured concentrations with regulatory limits, and a one-way ANOVA was performed to assess variations in heavy metal concentrations among regions. A post hoc multiple comparisons test is performed using the Games–Howell test for comparing the means among the various regions. Statistical significance was set at *p* ≤ 0.05. The percentage of samples exceeding the permissible limits was also calculated to evaluate the extent of contamination. 

## 3. Results and Discussion

### 3.1. Pb and Cd Contamination in the Lebanese Yogurt Samples

Significant regional variations were observed in the Pb and Cd concentrations in yogurt samples from various Lebanese regions, with some regions exceeding the regulation limitations set by the European Commission (EC) of 0.02 mg/kg and 0.005 mg/kg for Pb and Cd, respectively, in dairy products. The distribution of Pb and Cd values in each sampled region is shown in Figure 2 and Figure 3, emphasizing regional differences.

#### 3.1.1. Pb Contamination

Figure 2, a simple error bar of the mean graph, illustrates the significant difference in Pb concentrations between geographical areas. The highest levels of Pb contamination were found in Bekaa (0.0193 mg/kg), Baalbeck-Hermel (0.0211 mg/kg), and South Lebanon (0.0216 mg/kg), exceeding or closely approaching the regulation limit (0.02 mg/kg), indicated by the red line shown in the graph. These results of lead contamination are usually attributed to hotspots for pollution, which could be caused by contaminated animal feed, industrial emissions, or contaminated water supplies utilized in dairy production [18]. Darwish et al. [47] reported that Bekaa, including its governorate Baalbeck-Hermel, is a prime agricultural land. Up to 70% of the water resources available are consumed by the agricultural sector. The limited soil and water resources have been under growing stress due to intensive agriculture, urbanization, and industrial activities occurring in this region. Farmers there are compelled to utilize contaminated water because of the lack of water. They also mentioned that the excessive use of fertilizers and the presence of industrial wastes resulted in soil and groundwater contamination with heavy metals. Similarly, for South Lebanon, Borjac et al. [4] reported the water contamination with heavy metals and the concentrations were above the acceptable limits. In addition, they confirmed that the studied water sources are not safe for neither drinking nor irrigation. North Area (0.0170) and Nabatiyeh (0.0143 mg/kg) also exhibit high Pb concentrations; however, they remain below the acceptable limit. On the other hand, Pb levels were comparatively lower in areas like Metn (0.0047 mg/kg), Chouf (0.0073 mg/kg), and Akkar (0.0108 mg/kg), with the majority of samples staying below the detection limit. Pb contamination is not evenly distributed throughout Lebanon, as further evidenced by the statistical confirmation of the considerable regional variations in Pb concentrations (Table 2).

#### 3.1.2. Cd Contamination

In addition, Figure 3 illustrates the significant geographical fluctuation in Cd contamination, which was similar to that of Pb. North Area (0.01117 mg/kg), Baalbeck-Hermel (0.01024 mg/kg), and Bekaa (0.01024 mg/kg) had the highest Cd levels, surpassing the EC limit of 0.005 mg/kg by 2 to 3 times. These elevated concentrations are likely linked to environmental pollution. Cattle ingesting contaminated feed, water, and soil are the main ways that cadmium enters the dairy supply chain. It then bioaccumulates in the tissues of the cattle and eventually secretes into the milk. Cd migration in dairy products can also be affected by storage, including the type of container used, the period of storage, and exposure to the environment. Mining, industrial operations, and inadequate waste management all contribute to contamination by lowering the quality of the soil and water, which in turn affects the composition of milk and cattle feed [48]. This was confirmed by a case study done by Moussa et al., who reported the Cd contamination in the Bekaa valley, as it is a major agricultural area in Lebanon, leading to contamination in the soil and water [49]. For the North Area, the study by Saikali and El Azzi showed Cd contamination in the soil in Tannourine, a village located in the North Area of Lebanon. They detected Cd levels above the permitted level, and they attributed these results to different factors. This can be due to atmospheric deposition from industrial areas, agricultural practices leading to contamination of soils with persistent pesticides and phosphate fertilizers, erosion of bedrocks releasing Cd into the soil, or runoff from cultivated lands in the coastal areas [30]. Moderate contamination was observed in Nabatiyeh (0.00957 mg/kg), Batroun (0.00837 mg/kg), and Kesserwan (0.00526 mg/kg). Although the Cd levels in these regions exceeded the permissible limit, they were less than that exhibited by North Area, Baalbeck-Hermel, and Bekaa, suggesting a lower environmental contamination. On the other hand, the lowest levels of Cd were found in the yogurt samples from Akkar (0.00364 mg/kg), Jbeil (0.00424 mg/kg), Metn (0.00035 mg/kg), South Lebanon (0.00470 mg/kg), and Chouf (0.00397 mg/kg), falling significantly below the detection limit. Notably, no region remained fully within the regulatory threshold, highlighting the widespread presence of Cd in yogurt samples more than Pb. Table 2. highlight the widespread presence of Cd in yogurt samples with varying statistical significance.

There are few studies assessing the Pb and Cd levels in yogurts. More studies focused on their evaluation in milk [48,50,51,52,53,54] and other milk products, such as cheese [16,55], cream [48], and ghee [56]. Nevertheless, Pb and Cd concentrations can vary significantly between countries [57,58], which can be attributed to variances in industrial emissions, agricultural practices, and environmental contamination [59,60]. Table 3 shows a literature comparison for the variation in Pb and Cd contamination in yogurt samples across different countries, highlighting Lebanon’s position relative to other regions in the world. It was shown that Lebanese yogurt has modest levels of Pb (4.7–21.6 µg/kg) and Cd (3.5–12.77 µg/kg) in comparison to other countries. Pb levels in the Slovak Republic (260–360 µg/kg) are 12–76 times higher than those in Lebanon, whereas those in Pakistan (133–465 µg/kg) are up to 21 times higher. Pb levels are also much higher in Poland (up to 99 µg/kg) and Iran (52.19 µg/kg), while Pb contamination in Iraq (0.4–4.36 µg/kg) is up to 10 times lower. Similarly, Romania (5.93–6.16 µg/kg) and Libya (12 µg/kg) have similar amounts of Cd, while Brazil (35.4–210 µg/kg) and the Slovak Republic (50–70 µg/kg) have levels up to 16 times greater than Lebanon. Poland has far lower levels of Cd (0–6 µg/kg) than Iraq (0.02 µg/kg). The need for continuous monitoring is highlighted by the fact that, although Lebanon’s Pb levels are occasionally close to the EU limit of 20 µg/kg, its Cd contamination is still higher than acceptable levels in less polluted areas.

### 3.2. Public Health Implications

Table 4 presents the number and percentage of yogurt samples that exceeded the regulatory limits set by the EC to further evaluate the level of Pb and Cd contamination in the samples. Overall, 12.12% of the yogurt samples that were examined had Cd values over the 0.005 mg/kg standard, while 15.15% had Pb amounts above 0.02 mg/kg. These results show a high degree of heavy metal contamination in dairy products, which raises concerns about possible long-term health effects.

For both Pb (20%) and Cd (13.33%), South Lebanon, Baalbeck-Hermel, Jbeil, and Bekaa had the highest percentage of non-compliant samples among all regions. Localized contamination hotspots were highlighted by these regions’ higher average contamination levels as well as the increased number of samples that exceeded regulatory limits. On the other hand, the majority of samples fell within safe limits in areas like Metn and Chouf, which had the lowest contamination levels. Significant percentages of samples had Pb and Cd levels above safety limits, indicating the need for immediate action and more stringent monitoring in high-risk areas.

The findings in Table 4 highlight the critical need for intervention in the high-risk areas where pollution levels alarmingly exceed the acceptable limits. When Pb and Cd levels in yogurt samples exceed permissible limits, long-term public health concerns occur, particularly for vulnerable populations such as children, pregnant women, and those with chronic illnesses. Given the significant proportion of non-compliant samples in different regions, immediate action should be taken to decrease contamination sources and establish stricter quality control criteria.

### 3.3. One-Way ANOVA and Post Hoc Tests

Table 5 depicts the results of the one-way analysis of variance (ANOVA) results, which clearly shows that there is no significant difference between the means of Pb and Cd between the various regions where tests were performed, with significance level *p* = 0.876 for Pb and *p* = 0.853 for Cd.

Table 6 further confirms the previous results by using a post hoc Games–Howel multiple comparison test by creating 95% confidence intervals (CIs) of the difference between the mean concentrations of Cd and Pb of the various pairs of regions.

All CIs contain the value 0, confirming that there is no significant difference between the mean concentrations of Pb and Cd in the various regions.

### 3.4. Health Risk Characterization Based on Dietary Exposure to Pb and Cd Through Yogurt Consumption

The European Food Safety Authority (EFSA) established in 2009 a new tolerable weekly intake (TWI) value for Cd at 2.5 µg/kg b.w./week, which is significantly lower than the provisional tolerable weekly intake (PTWI) established by FAO/WHO in 1993 (7 µg/kg b.w./week). Several international bodies recognized that the margin between this PTWI and the actual weekly intake of cadmium by the general population was small and, in some populations, may be non-existent [69]. It became clear that the daily intake, even in agricultural and uncontaminated areas, may be close to the limit [70]. According to Hoteit et al. [46], the average body weight for Lebanese adult consumers is 73.8 kg; therefore, the daily intake of Cd with food should not exceed 24.35 µg. The obtained results shown in Table 7 reveal that the mean intake of Cd was 0.46 µg per day and accounts for only 1.89% of the calculated TDI. In a study done by Nasreddine et al. [45], it was reported that among the dairy category, yogurt was the most consumed dairy product (68.3 g/day), followed by cheese (35.1 g/day) and then by labneh (27.8 g/day), a traditional dairy product; thus, it accounted for 52% of the dairy products consumed. Furthermore, a study by Nasreddine et al. in 2010 [71] estimated the mean intake of Cd from all dairy products to be 0.44 µg per day, which is lower than the value found in the present study for yogurt alone.

This significant increase in Cd concentrations in dairy products over time may reflect rising environmental contamination, inadequate regulatory enforcement, and the persistent presence of pollutants in animal feed and water sources. These findings underscore the urgency of preventive measures, as Cd levels in foodstuffs and their overall dietary intake should be strictly controlled and continuously monitored, given its high toxicity and exceptionally long biological half-life, ranging from 10 to 30 years [69,72].

Foodstuffs are the main source of Cd exposure for the non-smoking general population. Cd absorption after dietary exposure in humans is relatively low (3–5%), but it is efficiently retained in the kidney and liver in the human body. Cd is primarily toxic to the kidney, especially to the proximal tubular cells, where it accumulates over time and may cause renal dysfunction. It can also cause bone demineralization, either through direct bone damage or indirectly as a result of renal dysfunction [69].

Due to these well-documented health risks, the European Commission (EC) has implemented strict regulations on cadmium levels in various food products [26]. Moreover, even if the exposure is lower than recommended by EFSA or FAO/WHO guidelines, distinct pathologies in many organ systems are observed, which supports efforts to reduce the permissible limits of cadmium intake and continuous monitoring of the content of this element in food [73].

Moreover, as in the case with Cd, in 2010, EFSA questioned the current value of PTWI for Pd (25 g/kg b.w.) as not providing complete safety for the population, as there was no evidence for a threshold for critical Pb-induced effects [74,75]. However, the new limit was not established. Instead, the CONTAM Panel identified a range of values for the 95% lower confidence limit of the benchmark dose of 1% extra risk (BMDL_01_) for each end point [70]. Using this model, BMDL dietary Pb intake values in adults of 1.50 µg/kg b.w. per day and 0.63 µg/kg b.w. per day were derived for the cardiovascular and kidney effects, respectively. A BMDL_01_ dietary intake value of 0.50 µg/kg b.w. per day for developmental neurotoxicity was derived [76]. The obtained results, as shown in Table 7, reveal that the mean intake of Pb through yogurt consumption is 0.94 µg per day, corresponding to 0.013 µg/kg body weight per day. While this value may appear relatively low, it is significantly higher when compared to the estimated mean daily intake of Pb through dairy consumption reported by Nasreddine et al. in 2010 [71], which was 0.08 µg per day. This indicates a tenfold increase in Pb intake from dairy sources over the past decade.

This striking rise in Pb intake through yogurt suggests a growing concern regarding environmental contamination, outdated agricultural practices, and possible lapses in food quality control. It highlights the urgent need for systematic monitoring of heavy metals in dairy products and the implementation of effective regulatory measures to mitigate public health risks. Due to its long half-life in the body, chronic toxicity of Pb is of most concern when considering the potential risk to human health. In humans, the central nervous system is the main target organ for Pb toxicity. In adults, Pb-associated neurotoxicity was found to affect central information processing, especially for visuospatial organization and short-term verbal memory, to cause psychiatric symptoms, and to impair manual dexterity. There is considerable evidence demonstrating that the developing brain is more vulnerable to the neurotoxicity of Pb than the mature brain. In children, an elevated blood lead level is inversely associated with a reduced intelligence quotient (IQ) score and reduced cognitive functions up to at least seven years of age [75].

Therefore, since food is the major source of human exposure to Pb and there is no recommended tolerable intake level, the Pb content in food products should be kept as low as possible [76,77].

## 4. Safety Policy Recommendation for Dairy Products

The safety of dairy products necessitates addressing contamination threats on several levels. The most important thing is to reduce heavy metal contamination. To decrease the incidence of contamination in dairy products, a multi-step approach that includes source identification, detection, and regulatory enforcement is needed. Pb and Cd can enter dairy products via dirty soil, water, and animal feed; therefore, environmental control is an important initial step. Limiting industrial emissions, enforcing soil remediation in damaged agricultural zones, and regulating livestock feed quality can all help to reduce these metal levels in milk, which is the origin of all dairy products. The first step for this approach is the requirement of regular testing and enhanced detection methods. Additionally, advanced analytical techniques should also be incorporated into routine monitoring programs to detect contamination in the early stages. Establishing checkpoints at several stages, from milk collection to final processing, guarantees that contaminated samples are detected and eliminated before they reach consumers. The second step is the education of dairy farmers, who must be instructed on optimal procedures for reducing heavy metal absorption by livestock. Adjusting feed composition, monitoring water sources, and using sustainable farming techniques can all help to decrease contamination risks. Last but not least, regulatory authorities should require stringent compliance with heavy metal limitations in dairy products, such as applying routine inspections, standardized testing processes, and penalties.

## 5. Conclusions

The Pb and Cd contamination levels in yogurt samples from several Lebanese regions are evaluated in this study. The findings show that a significant percentage of the samples had levels of Pb (10.9%) and Cd (14.5%) surpassing the regulatory limits, with North Area, Baalbeck-Hermel, and South Lebanon having the highest levels of contamination. The detection of elevated potentially toxic elements/heavy metals concentrations in several regions raises concerns about consumer health risks and regulatory compliance. The regional differences in heavy metal concentrations point to a variety of contamination sources, including industrial emissions, agricultural activities, and environmental pollution. In addition, these results highlight the necessity of more stringent regulatory enforcement and regular monitoring to ensure the safety of dairy products in Lebanon. Exposure hazards can be considerably decreased by addressing contamination at its source through strict quality control procedures, better wastewater management, and enhanced farming practices. Future research should focus on the identification of the origins of contamination and evaluating the long-term effects of consuming dairy products that expose consumers to heavy metals.

## Figures and Tables

**Figure 1 toxics-13-00499-f001:**
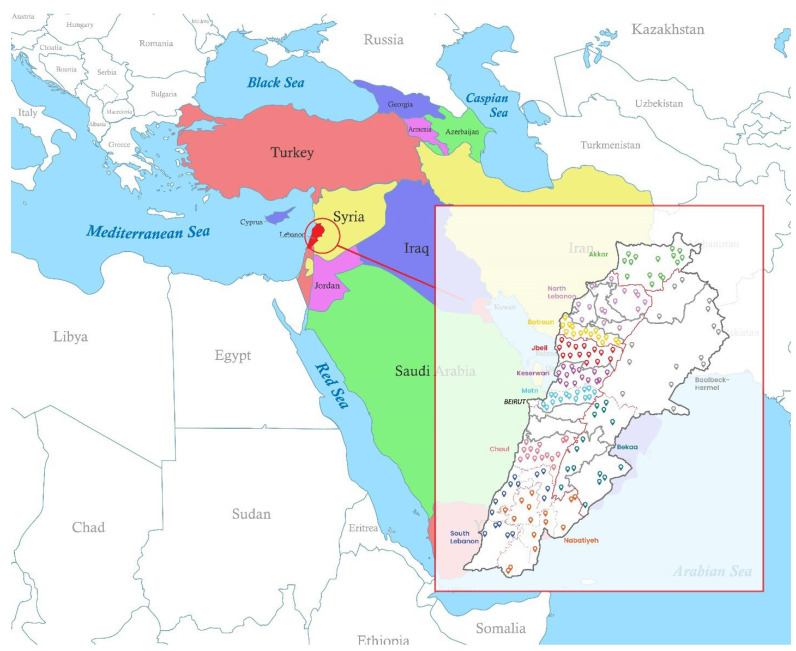
View of Lebanon’s location on the global map with the geographic distribution of the yogurt sampling sites across the eleven Lebanese regions.

**Figure 2 toxics-13-00499-f002:**
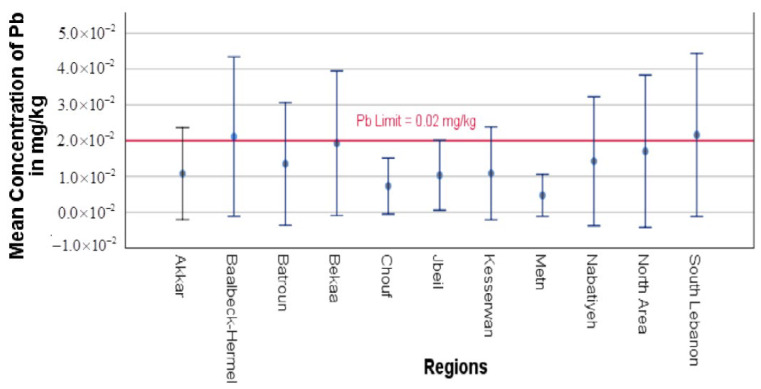
Simple error bar of the mean graph of Pb concentrations across different Lebanese regions. Error bars represent 95% confidence intervals of the raw data.

**Figure 3 toxics-13-00499-f003:**
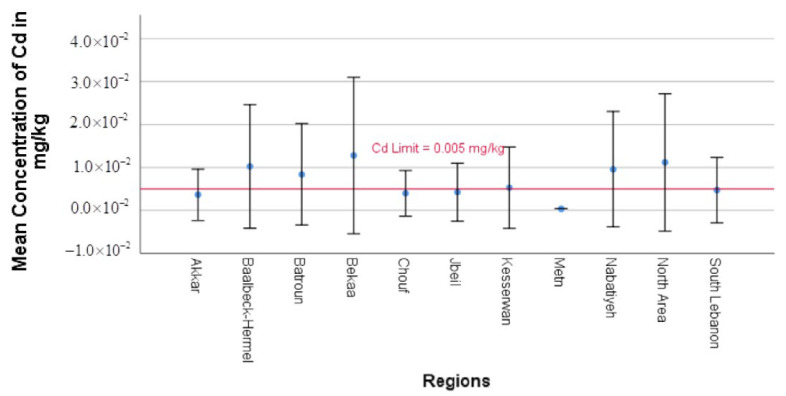
Simple error bar of the mean graph of Cd concentrations across different Lebanese regions. Error bars represent 95% confidence intervals of the raw data.

**Table 1 toxics-13-00499-t001:** Average nutritional composition of the collected yogurt samples.

Component	Average Content per 100 g
Fat	3.5 g
Protein	3.3 g
Carbohydrates	4.5 g
Energy	65 kcal
Additives	None (artisanal/natural)
Milk Source	100% regional cow’s milk

**Table 2 toxics-13-00499-t002:** One-sample two-sided *t* test and 95% confidence intervals (CIs) for the mean concentrations of Pb and Cd. Table 2 a. Concentrations using the raw data. Table 2 b. Concentrations using bootstrapping with 1000 resampled samples distribution.

Table 2 a Concentrations *	Pb Concentrations (Limit = 0.02 mg/kg)			Cd Concentrations (Limit = 0.005 mg/kg)		
	*p*-Value	Mean	95% CI	*p*-Value	Mean	95% CI
Akkar	0.147	0.0108	–0.0021—0.0237	0.632	0.00364	–0.00234—0.00961
Baalbeck-Hermel	0.915	0.0211	–0.0011—0.0434	0.450	0.01024	–0.00422—0.02469
Batroun	0.432	0.0135	–0.0036—0.0307	0.551	0.00837	–0.00346—0.02020
Bekaa	0.939	0.0193	–0.0009—0.0395	0.376	0.01277	–0.00547—0.03101
Chouf	0.004	0.0073	–0.0005—0.0152	0.684	0.00397	–0.00135—0.00929
Jbeil	0.052	0.0103	0.0006—0.0201	0.812	0.00424	–0.00252—0.01100
Kesserwan	0.153	0.0109	–0.0021—0.0238	0.953	0.00526	–0.00425—0.01477
Metn	<0.001	0.0047	–0.0011—0.0106	<0.001	0.00035	0.00035—0.00035
Nabatiyeh	0.506	0.0143	–0.0038—0.0323	0.479	0.00957	–0.00391—0.02305
North Area	0.770	0.0170	–0.0042—0.0383	0.423	0.01117	–0.00486—0.02720
South Lebanon	0.882	0.0216	–0.0012—0.0444	0.935	0.00470	–0.00291—0.01232
All Regions	0.007	0.0137	0.0092—0.0182	0.260	0.00675	0.00369—0.00982
**Table 2 b Concentrations ****	**Pb Concentrations (Limit = 0.02 mg/kg)**			**Cd Concentrations (Limit = 0.005 mg/kg)**		
	***p*-Value**	**Mean**	**95% CI**	***p*-Value**	**Mean**	**95% CI**
Akkar	0.124	0.0108	0.0020—0.0241	0.592	0.00364	0.00035—0.00970
Baalbeck-Hermel	0.895	0.0211	0.0075—0.0443	0.501	0.01024	0.00035—0.02517
Batroun	0.368	0.0135	0.0020—0.0318	0.599	0.00837	0.00035—0.01839
Bekaa	0.907	0.0193	0.0067—0.0390	0.349	0.01277	0.00035—0.03083
Chouf	0.072	0.0073	0.0020—0.0156	0.713	0.00397	0.00035—0.00957
Jbeil	0.070	0.0103	0.0041—0.0195	0.834	0.00424	0.00035—0.01046
Kesserwan	0.115	0.0109	0.0020—0.0244	0.853	0.00526	0.00035—0.01415
Metn	0.231	0.0047	0.0020—0.0102	0.002	0.00035	0.00035—0.00035
Nabatiyeh	0.517	0.0143	0.0020—0.0321	0.491	0.00957	0.00035—0.02350
North Area	0.711	0.0170	0.0020—0.0391	0.342	0.01117	0.00035—0.02657
South Lebanon	0.865	0.0216	0.0073—0.0435	0.850	0.00470	0.00035—0.01257
All Regions	0.016	0.0137	0.0092—0.0185	0.246	0.00675	0.00387—0.01036

* Concentrations using the raw data.** Concentrations using bootstrapping with 1000 resampled samples distribution.

**Table 3 toxics-13-00499-t003:** Literature comparison of Pb and Cd contamination in yogurt samples across different countries.

Country	Pb (µg/kg)	Cd (µg/kg)	Ref.
Iran	52.19	1.21	[48]
Brazil	21.58	4.20	[30]
Brazil	2.5–12.4	35.4–210	[61]
Brazil	20	5	[62]
Pakistan	133–465	15–59	[63]
Libya	20	12	[64]
Iraq	0.4–4.36	0.02	[65]
Poland	4–99	0–6	[66]
Romania	9.88–11.30	5.93–6.16	[67]
Slovak Republic	260–360	50–70	[68]
Lebanon	4.7–21.6	3.5–12.77	This study

**Table 4 toxics-13-00499-t004:** Number and percentage of yogurt samples exceeding regulatory limits for Pb and Cd.

	Pb Concentrations (Limit = 0.02 mg/kg)			Cd Concentrations (Limit = 0.005 mg/kg)		
	Total Number of Samples	Number of Samples Above the Limit	Percentage of Samples above the Limit	Total Number of Samples	Number of Samples Above the Limit	Percentage of Samples above the Limit
Akkar	15	2	13.33	15	2	13.33
Baalbeck-Hermel	15	3	20.00	15	2	13.33
Batroun	15	2	13.33	15	2	13.33
Bekaa	15	3	20.00	15	2	13.33
Chouf	15	2	13.33	15	2	13.33
Jbeil	15	3	20.00	15	2	13.33
Kesserwan	15	2	13.33	15	2	13.33
Metn	15	1	6.67	15	0	0.00
Nabatiyeh	15	2	13.33	15	2	13.33
North Area	15	2	13.33	15	2	13.33
South Lebanon	15	3	20.00	15	2	13.33
All Regions	165	25	15.15	165	20	12.12

**Table 5 toxics-13-00499-t005:** One-way analysis of variance (ANOVA) for Pb and Cd between the various regions.

		Sum of Squares	Degrees of Freedom	Mean Square	F	*p*-Value
Pb	Between Groups	0.005	10	0.000	0.518	0.876
	Within Groups	0.138	154	0.001		
	Total	0.142	164			
Cd	Between Groups	0.002	10	0.000	0.549	0.853
	Within Groups	0.063	154	0.000		
	Total	0.065	164			

**Table 6 toxics-13-00499-t006:** Post hoc Games–Howell and multiple comparison tests for Pb and Cd concentrations between all the various pairs of regions; 95% CI of the difference between pairs of regions.

Cd and Pb	95% Confidence Interval for the Difference of Concentrations Between all Pairs of Regions										
	Akkar	Baalbeck-Hermel	Batroun	Bekaa	Chouf	Jbeil	Kesserwan	Metn	Nabatiyeh	North Area	South Lebanon
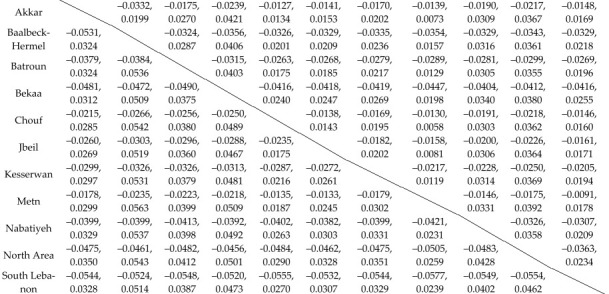											

Above the table dividing line is Cd, and below the table dividing line is Pb.

**Table 7 toxics-13-00499-t007:** Daily intake of Cd and Pd.

	Cd Daily Intake (µg/day)	Cd Daily Intake (µg/kg b.w./day)	Pb Daily Intake (µg/day)	Pd Daily Intake (µg/kg b.w./day)
Akkar	0.25	0.003	0.74	0.010
Baalbeck-Hermel	0.70	0.009	1.44	0.020
Batroun	0.57	0.008	0.92	0.012
Bekaa	0.87	0.012	1.32	0.018
Chouf	0.27	0.004	0.50	0.007
Jbeil	0.29	0.004	0.70	0.010
Kesserwan	0.36	0.005	0.74	0.010
Metn	<LOQ	<LOQ	0.32	0.004
Nabatiyeh	0.65	0.009	0.98	0.013
North Area	0.76	0.010	1.16	0.016
South Lebanon	0.32	0.004	1.48	0.020
Mean Value	0.46	0.006	0.94	0.013
TDI [49]	24.35	0.36		

## Data Availability

The data are contained within the article. Additional data are available upon request from the corresponding authors.

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
