# Peer review of "Heavy Metal Contamination in Yogurt from Lebanon: Evaluating Lead (Pb) and Cadmium (Cd) Concentrations Across Multiple Regions"

_toxics, 2025, doi:10.3390/toxics13060499_

Round 1
Reviewer 1 Report
Comments and Suggestions for Authors
The paper presents the content of lead and cadmium in yogurt from Lebanon. Some parts should be significantly changed.
Update the ref. [20].
Commission Regulation (EU) 2023/915 of 25 April 2023 on maximum levels for certain contaminants in food and repealing Regulation (EC) No 1881/2006 (Text with EEA relevance).
Is the European Union standard applicable in Lebanon?
Citations of other studies should be as in ref [3] – line 52, this applies to the all manuscript, the surname et al. [ ]
The spelling of Pb and Cd should be standardized, abbreviations should be used, the names of elements should be written in lower case
Introduction
Line 63: Please start a new paragraph
- Materials and Methods
2.1. Sampling
This section should be significantly expanded, explaining what types of yogurts these were, whether they were natural, whether they contained additives, with what content of fat, protein, carbohydrates. Present this information in a table. This may be important especially in the matter of cadmium content. There is no information whether yogurts from a given region were purchased or produced there, whether the milk came from the region.
2.3. Preparation of Samples
Was it necessary to filter after mineralization, for what purpose was it carried out?
- Results and Discussion
Fig. 2 and 3 no unit.
Fig. 3 the numerical scale is unreadable, remove 2 zeros
Did the price of yogurt affect the quality and content of Cd and Pb?
References
Ref. 13, 14, 23, 47 spelling should be corrected.
Author Response
Comment 1: Update the ref. [20].
Commission Regulation (EU) 2023/915 of 25 April 2023 on maximum levels for certain contaminants in food and repealing Regulation (EC) No 1881/2006 (Text with EEA relevance).
Is the European Union standard applicable in Lebanon?
Response 1: hank you for bringing this to our attention. We have updated Reference [20], now [26], in the manuscript’s reference list (page 32, line 755), replacing the previous citation:
Union, P.O. of the E. (2021). C/2021/5831, Commission Regulation (EU) 2021/1317 of 9 August 2021 amending Regulation (EC) No 1881/2006 as regards maximum levels of lead in certain foodstuffs. (Text with EEA relevance), https://op.europa.eu/en/publication-detail/-/publication/f036600e-f974-11eb-b520-01aa75ed71a1/language-en
with the updated regulation:
Commission Regulation (EU) 2023/915 of 25 April 2023 on maximum levels for certain contaminants in food and repealing Regulation (EC) No 1881/2006 (Text with EEA relevance).
In addition, the corresponding citation in the manuscript text (page 4, line 140) was revised to:
According to Commission Regulation (EU) 2023/915, which repeals Regulation (EC) No 1881/2006, raw milk and dairy products may contain up to 0.020 mg/kg of lead and 0.005 mg/kg of cadmium, respectively [20].
All relevant modifications have been marked in red in the revised manuscript.
As for the applicability of the European standard, it is important to note that Lebanon applies national standards issued by the Lebanese Standards Institution (LIBNOR) for products placed on the local market. In the case of yogurt, the relevant standard is LS 33:1999 – Fermented Yogurt, which outlines general product specifications but does not include maximum residue limits (MRLs) for heavy metals such as lead (Pb) and cadmium (Cd).
In situations where local standards lack specific guidance, it is common and accepted practice to refer to internationally recognized benchmarks, particularly those established by the European Commission, which are widely used in food safety risk assessment and regulatory comparison. Accordingly, in our study, we adopted the European Union limits for Pb and Cd as reference values to evaluate contamination levels in yogurt samples and to allow for consistency with international research.
Comment 2: Citations of other studies should be as in ref [3] – line 52, this applies to the all manuscript, the surname et al. [ ]
The spelling of Pb and Cd should be standardized, abbreviations should be used, the names of elements should be written in lower case
Response 2: We agree with the comment. All in-text citations have been revised to follow the format “Surname et al. [X]” as exemplified in reference [3], and this correction has been applied consistently throughout the manuscript. Additionally, the abbreviations Pb and Cd have been standardized, and all element names are now written in lowercase in accordance with scientific conventions. All relevant changes have been highlighted in red in the revised manuscript.
Comment 3: Introduction Line 63: Please start a new paragraph
Response 3: Thank you for the observation. A new paragraph has been initiated in page 2 at Line 80, beginning with the sentence starting “The impact…” as suggested. This adjustment has been clearly marked in red in the revised manuscript.
Comment 4: 2. Materials and Methods 2.1. Sampling
This section should be significantly expanded, explaining what types of yogurts these were, whether they were natural, whether they contained additives, with what content of fat, protein, carbohydrates. Present this information in a table. This may be important especially in the matter of cadmium content. There is no information whether yogurts from a given region were purchased or produced there, whether the milk came from the region.
Response 4: We thank you for this insightful comment. In response, we have expanded the “Sampling” section to include all the requested details regarding the type, origin, and composition of the yogurt samples. The following paragraph was added to the manuscript in pages 5-6 from line 185 to line 204 (marked in red):
Yogurt samples were collected from 11 different Lebanese regions: Nabatiyeh (Na), South Area (S), Chouf (C), Metn (M), Kesserwan (K), Jbeil (J), Batroun (B), North Area (No), Akkar (A), Bekaa (Be), and Baalbeck-Hermel (BH). A total of 15 samples were obtained from different dairy farms and processing plants in each region, resulting in 165 samples. All yogurt products were produced using traditional methods, within the farm’s premises without the addition of preservatives, stabilizers, or flavoring agents, and were made exclusively from cow’s milk sourced from the same region of production. These yogurts reflect the naturally processed dairy products commonly consumed in Lebanon and represent the regional diversity of local milk production practices
Following collection, the samples were immediately transported under refrigerated and sterile conditions to the laboratory facilities. Upon arrival, the samples were subjected to chemical analysis without delay, in order to preserve their integrity and prevent any potential degradation or alteration in contaminant concentrations. The general nutritional composition of the collected yogurt samples is summarized in Table 1. The geographic distribution of the study areas is illustrated in Figure 1.
Table 1: Average Nutritional Composition of the collected Yogurt Samples
|
Component |
Average Content per 100 g |
|
Fat |
3.5 g |
|
Protein |
3.3 g |
|
Carbohydrates |
4.5 g |
|
Energy |
65 kcal |
|
Additives |
None (artisanal/natural) |
|
Milk Source |
100% regional cow’s milk |
Comment 5: 2.3. Preparation of Samples
Was it necessary to filter after mineralization, for what purpose was it carried out?
Response 5: Thank you for your valuable observation. Filtration after mineralization was included in our procedure to ensure the removal of any remaining undissolved particulate matter, even after complete digestion. Although microwave-assisted digestion typically results in a clear solution, minute residues—especially from fat or protein matrices in dairy products like yogurt—can still persist and may clog the nebulizer or affect the precision of the instrumental analysis (in our case, atomic absorption spectrometry).
The use of 4 µm PTFE filters specifically serves to:
- Protect the analytical instrument (e.g., the graphite furnace or flame atomizer in AAS) from damage or contamination,
- Prevent baseline drift or matrix interference caused by particulates,
- Ensure reproducibility and accuracy of metal concentration measurements, particularly at trace levels.
This step is in accordance with standard practice in trace metal analysis and was also employed in the validated method we adapted from Kassouf et al. (2013), which emphasizes the importance of a clean, particulate-free digest for optimal analytical performance.
Comment 6: 3. Results and Discussion
Fig. 2 and 3 no unit.
Fig. 3 the numerical scale is unreadable, remove 2 zeros
Did the price of yogurt affect the quality and content of Cd and Pb?
Response 6: Thank you for your observation. The type of graph used for Figures 2 and 3 has been changed to improve clarity. Units and axis labels have been added accordingly, and the numerical scale in Figure 3 was adjusted to enhance readability. Regarding the price of yogurt, Heavy metal content is not conclusively established by production cost.
Comment 7: References
Ref. 13, 14, 23, 47 spelling should be corrected.
Response 7: Thank you for your observation. The spelling of the references has been corrected as requested. Please note that due to adjustments in the reference list, the numbering has changed as follows:
- Reference 13 is now 19 (line 733, page 22)
Bousbia, A., Boudalia, S., Gueroui, Y., Ghebache, R., Amrouchi, M., Belase, B., Meguelati, S., Belkheir, B., Benidir, M., Chelaghmia, M.L.: Heavy metals concentrations in raw cow milk produced in the different livestock farming types in Guelma province (Algeria): Contamination and risk assessment of consumption. Journal of Animal and plant Sciences, 2019, 29, 2, 386, EBSCOhost, https://openurl.ebsco.com/contentitem/gcd:136245142?sid=ebsco:plink:crawler&id=ebsco:gcd:136245142 - Reference 14 is now 20(line 738, page 23)
Mahmoudi, R., Kazeminia, M., Kaboudari, A., Mahalleh, S.P.-, Pakbin, B.: A review of the importance, detection and controlling of heavy metals in milk and dairy products, MJS., 2017, 36, 1-6, https://doi.org/10.22452/mjs.vol36no1.1 - Reference 23 is now 29 (line 763, page 23)
Yi-Jie Dai; Alsayeqh, Abdullah F.; Ali, Eman Wagih E. E.; Abdelaziz, Ahmed S.; Khalifa, Hesham A.; Mohamed, Asmaa S. M.; Alnakip, Mohamed E.: Heavy metals content in cheese: A study of their dietary intake and health risk assessment. Slovenian Veterinary Research, 2023, 60, 397, https://doi.org/10.26873/SVR-1639-2022 - Reference 47 is now 67 (line 865, page 25)
Prioteasa, L., Prodana, M.: Evaluation of heavy metals and toxic elements content in time in bio and non-bio yogurts by ICP-MS method, UPB Scientific Bulletin, Series B, 2014, 76, 141-150.
All changes have been updated in the reference list and highlighted in red in the revised manuscript.
Reviewer 2 Report
Comments and Suggestions for Authors
I would suggest that the authors revise the nomenclature to refer to the elements Pb and Cd (e.g. potentially toxic elements), since Pb has no physiological function in the human organism and is toxic even in small concentrations.
Introduction: I understand that Pb and Cd are toxic elements and therefore pose risks to human health. However, why did the authors choose to quantify only these two elements in the yogurt samples? Is there an important source of Pb and Cd in the study area? For example, have any studies reported worrying concentrations of Pb and Cd in other environmental matrices (e.g. soil, vegetation)? Do all the collection areas have the same potential sources of pollution? Does the contamination occur during milk production (i.e. bioaccumulation in cows) or during yogurt manufacture? The motivation for the study is unclear.
L46-80: This paragraph is excessively long, making it difficult to read. I suggest dividing this paragraph into at least two new paragraphs.
L107-108: “This study aims to evaluate the levels of Pb and Cd in Lebanese yogurt samples collected from various regions and assess their compliance with regulatory limit”. The objectives denote a purely descriptive study - and indeed it is. I suggest that the authors calculate the estimated intake of the toxic metals through yogurt consumption, broadening the discussion about the risks to human health rather than just quantifying the concentrations of the toxic elements.
Section 2.1: Are there any differences between the collection sites that should be noted? For example, potential sources of pollution in certain areas, in order to justify the differences in concentrations found in the present study. I think it would be important to include a sub-section in the M&M for a brief description of the study areas in relation to demographics, economy and potential pollution sources.
Figure 1: Not all readers are familiar with the geographical location of Lebanon. Authors should also represent the location in relation to the continent and the world.
Section 2.5: Specify the detection and quantification limits of the analytical method for both elements.
Section 2.6: The authors used only parametric tests. Which statistical test was used to assess the distribution of the data? Did all the subsets of data (i.e. samples/location) have a normal distribution? I think it's very unlikely. Additionally, did the authors use a post hoc multiple comparison test to check which locations differed from each other?
Figures 2 and 3: The graphs are not boxplots. Indeed, the graph used is not recommended, as it is not possible to see the variability of concentrations in each location. The authors should replace these graphs with real boxplots. In addition, the statistical differences determined by the comparison tests should be represented in the graphs (e.g. letters indicating different groups).
Section 2.6, Tables 1 and 3: It is not clear which groups were compared in the ANOVA: (i) sample units (triplicates) in each location; or (ii) concentrations between locations. If it is the second case (ii) - which makes more sense - there should not be a p-value for each location. On the other hand, the comparison within the same locality (i) does not make sense and does not demonstrate significant information. Please review the data analysis method, including clarification on multiple comparisons after ANOVA.
Author Response
Comment 1: I would suggest that the authors revise the nomenclature to refer to the elements Pb and Cd (e.g. potentially toxic elements), since Pb has no physiological function in the human organism and is toxic even in small concentrations.
Response 1: Thank you for your valuable suggestion. We have revised the nomenclature throughout the manuscript to refer to Pb and Cd as potentially toxic elements (PTEs), as recommended. This terminology has been incorporated in lines 62, 86, 127, 138, and 649 of the revised version. All modifications have been highlighted in red.
Comment 2: Introduction: I understand that Pb and Cd are toxic elements and therefore pose risks to human health. However, why did the authors choose to quantify only these two elements in the yogurt samples? Is there an important source of Pb and Cd in the study area? For example, have any studies reported worrying concentrations of Pb and Cd in other environmental matrices (e.g. soil, vegetation)? Do all the collection areas have the same potential sources of pollution? Does the contamination occur during milk production (i.e. bioaccumulation in cows) or during yogurt manufacture? The motivation for the study is unclear.
Response 2: Thank you for your insightful comment. We agree that clarifying the motivation for selecting Pb and Cd is important. These elements were chosen based on several environmental assessments in Lebanon that have reported concerning levels of Pb and Cd in various matrices such as soil, groundwater, and vegetation—especially in regions included in our sampling. These pollutants are largely attributed to industrial emissions, overuse of fertilizers and pesticides, uncontrolled waste dumping, and contaminated water or feed used in livestock farming. For instance, Moustafa et al. [3] reported Cd and Pb as the heaviest pollutants in soil and groundwater in Akkar, North Lebanon. Borjac et al. [4] found exceedances of international limits for these elements in soils sampled near a dumpsite in Deir Kanoun, South Lebanon. Additionally, Darwish et al. [5] showed that ecological risk in the Central Bekaa Valley was predominantly driven by high to moderate levels of Cd and Pb. These findings indicate that the contamination is likely occurring during milk production through bioaccumulation in cows due to environmental exposure, rather than during yogurt manufacture. We have clarified this motivation and rationale in the introduction section of the revised manuscript in a paragraph highlighted in red from line 65 till line 79.
The newly added paragraph is highlighted in red in the revised manuscript.
Comment 3: L46-80: This paragraph is excessively long, making it difficult to read. I suggest dividing this paragraph into at least two new paragraphs.
Response 3: Thank you for your comment. We agree that the original paragraph was too long and could impact readability. In the revised manuscript, the section corresponding to the original lines 46–80 (now lines 59–125) has been significantly expanded with additional supporting information and has been divided into five distinct paragraphs to enhance clarity and readability.
Comment 4: L107-108: “This study aims to evaluate the levels of Pb and Cd in Lebanese yogurt samples collected from various regions and assess their compliance with regulatory limit”. The objectives denote a purely descriptive study - and indeed it is. I suggest that the authors calculate the estimated intake of the toxic metals through yogurt consumption, broadening the discussion about the risks to human health rather than just quantifying the concentrations of the toxic elements.
Response 4: We thank the reviewer for this valuable suggestion. In response, an Estimated Daily Intake (EDI) analysis was incorporated into the study. This addition appears in the Materials and Methods section under paragraph 2.8, entitled “Estimated Dietary Intake for Pb and Cd” in page 9-10. Furthermore, the corresponding findings and health implications are discussed in the Results and Discussion section under paragraph 3.4, currently titled “Health risk characterization based on dietary exposure to Pb and Cd through yogurt consumption” in page 17-18-19.
The newly added subsections are highlighted in red in the revised manuscript.
Comment 5: Section 2.1: Are there any differences between the collection sites that should be noted? For example, potential sources of pollution in certain areas, in order to justify the differences in concentrations found in the present study. I think it would be important to include a sub-section in the M&M for a brief description of the study areas in relation to demographics, economy and potential pollution sources.
Response 5: We thank the reviewer for this valuable suggestion. In response, a dedicated subsection titled “2.3. Description of the Study Areas” has been added to the Materials and Methods section (page 7). This section outlines the main economic activities and identifies potential pollution sources in each of the yogurt collection regions. While specific demographic data was not available, the added content helps contextualize the possible environmental factors contributing to variations in heavy metal concentrations across regions.
The newly added subsection is highlighted in red in the revised manuscript.
Comment 6: Figure 1: Not all readers are familiar with the geographical location of Lebanon. Authors should also represent the location in relation to the continent and the world.
Response 6: Thank you for the comment. In response, we have added a subsection in the Materials and Methods section entitled “2.1. Lebanon” in page 5 FROM LINE 170 TILL LINE 183 of the revised manuscript, which provides a brief description of the country's geographic location and general background. This aims to offer sufficient contextual information for international readers unfamiliar with Lebanon’s position globally.
The newly added paragraph is highlighted in red in the revised manuscript.
Comment 7: Section 2.5: Specify the detection and quantification limits of the analytical method for both elements.
Response 7: Thank you for your comment. The limits of detection (LODs) and limits of quantification (LOQs) for both Pb and Cd have been added to the revised manuscript under the updated subsection “2.7. Method Validation” in pages 9 (lines 327-330) of the revised manuscript. Specifically, the LODs for Pb and Cd were 0.004 mg/kg and 0.0007 mg/kg, respectively, and the LOQs were 0.012 mg/kg and 0.0021 mg/kg, respectively.
The newly added information is highlighted in red in the revised manuscript.
Comment 8: Section 2.6: The authors used only parametric tests. Which statistical test was used to assess the distribution of the data? Did all the subsets of data (i.e. samples/location) have a normal distribution? I think it's very unlikely. Additionally, did the authors use a post hoc multiple comparison test to check which locations differed from each other?
Response 8: We appreciate the reviewer’s observation regarding the distribution of the data. The reviewer is correct in that the data are not normally distributed, since the number of samples/location are/is not large enough. To overcome this issue, we performed additional tests using Bootstrapping, which involves repeatedly resampling the original data to create a new resampled distribution which allows us to create a better estimate of the Confidence Interval of the Mean and assess the significance level of the results.
Also, as the variances of the samples are unequal, post hoc multiple comparison test is performed using the Games-Howell test, which showed no significant difference between the levels of Cd and Pb at the various locations.
Comment 9: Figures 2 and 3: The graphs are not boxplots. Indeed, the graph used is not recommended, as it is not possible to see the variability of concentrations in each location. The authors should replace these graphs with real boxplots. In addition, the statistical differences determined by the comparison tests should be represented in the graphs (e.g. letters indicating different groups).
Response 9: Thank you for your comment. The graphs are indeed not boxplots, but rather bar graphs, and are wrongly captioned. But since the number of data points in the samples is not large enough a boxplot will not show the features and variability of the data. Instead, Simple Error Bar of the Mean graphs are used now in figure 2 and 3 which show the Mean and error bars representing 95% Confidence Intervals.
Comment 10: Section 2.6, Tables 1 and 3: It is not clear which groups were compared in the ANOVA: (i) sample units (triplicates) in each location; or (ii) concentrations between locations. If it is the second case (ii) - which makes more sense - there should not be a p-value for each location. On the other hand, the comparison within the same locality (i) does not make sense and does not demonstrate significant information. Please review the data analysis method, including clarification on multiple comparisons after ANOVA
Response 10: Thank you for pointing this out. Indeed, also wrongly captioned, Tables 2 (2.a & 2.b) and 4 actually represent One Sample 2-sided t test and 95% Confidence Intervals for the Mean, and Number and percentage of yogurt samples exceeding regulatory limits for Pb and Cd, respectively.
One Way ANOVA test is now added with post hoc multiple comparison test is performed using the Games-Howell test, which showed no significant difference between the levels of Cd and Pb at the various locations.
Reviewer 3 Report
Comments and Suggestions for Authors
General comment:
The manuscript represents practical study with the main aims to measure the levels of Pb and Cd in yogurt samples that were collected from 11 distinct Lebanese locations and determine their compliance with the legal limits set by the European Commission. This is well grounded concept, and the main aim has been well addressed through the manuscript. It brings enough novelty as no previous studies have been conducted to assess the presence of toxic metals, in particular, Pb and Cd, in yogurt samples across different regions in Lebanon.
The authors have well composed the study to be comprehensive enough in terms of analysing a total number of 165 yogurt samples (15 per each region) using atomic absorption spectrophotometry (AAS). The methodology has been well chosen and presented (sampling, reagents, preparation of samples, metal analysis, method validation and data analysis) and is robust enough to prove the findings are relevant. Most recent EU standards were used for comparation.
The study has been well presented in enough details. Interpretation of the results is containing relevant tables and figures that allow for good understanding of the research concept and steps taken. Results and discussion section is covering all the important aspects and is logically organised in two chapters related to lead and cadmium contamination. This is followed by safety policy recommendation for dairy products by the authors at levels of quality control procedures, better wastewater management, and enhanced farming practices.
Authors' statements are well supported with relevant references. Still, all of them are from other authors. The results are well presented in enough details. Conclusions are practical, and they are adequately reflecting the main point – regional differences in contamination, which are probably caused by industrial emissions, agricultural activities, and environmental pollution.
The findings of this study provide valuable insights into the potential health risks associated with yogurt consumption in Lebanon, as some regions are exceeding the regulation limitations set by the European Commission. This highlights the critical need for intervention in the high-risk areas where pollution levels alarmingly exceed the acceptable limits.
I consider the manuscript well written. It is practically useful, and I suggest accepting the manuscript for publication after addressing minor comments listed down.
Specific comments:
6-26 No need to write words “Affiliation 1, 2, 3…”.
31 Avoid starting sentences with numbers.
42 Delete green marks.
93 Delete “According to”. “Replaces” instead of “changes”.
95 “Cadmium” shouldn’t be capitalized.
97 Should stay “Er et al.”.
146, 240, 254, 256, 259 Delete yellow marks.
159 Should be italic.
183 Word “excellent” to be replaced with something more appropriate and in line with the text that is following it.
236 Should stay “There is little literature that has assessed lead and cadmium levels in yogurts.”.
Table 3. Capital letters only for first words in sentences.
323 Should not be bolded.
324-450 Some references from the authors should be added to prove previous experience, if possible.
Author Response
Comment 1: 6-26 No need to write words “Affiliation 1, 2, 3…”.
Response 1: Thank you for your observation. The words “Affiliation 1, 2, 3…” have been removed from the manuscript as requested. The affiliations have been amended as follows:
1 Saint Joseph University of Beirut (USJ), Faculty of Pharmacy, Department of Nutrition, Ecole Doctorale Sciences et Santé (EDSS), Medical Sciences Campus, Lebanon, Sandra.sarkis@net.usj.edu.lb
2 IMT Atlantique, Energy Systems and Environment Department, GEPEA, Rue Alfred Kastler, France, rayan.kashmar@imt-atlantique.fr
3 Doctoral School of Science and Technology, Research and Analysis Platform in Environmental Sciences (EDST-PRASE), Beirut, Lebanon, rayan.kashmar.1@ul.edu.lb
4 Faculty of Public Health, Charisma University, London EC1V 7QE, UK, nicolas@trccolleges.com
5 Institut National de Santé Publique, d’Epidémiologie Clinique et de Toxicologie-Liban (INSPECT-LB), Beirut, Lebanon, m.hoteit@ul.edu.lb
6 Food Sciences Unit, National Council for Scientific Research-Lebanon (CNRS-L), Beirut P.O. Box 11-8281, Lebanon.
7 PHENOL Research Group (Public Health Nutrition Program-Lebanon), Faculty of Public Health, Lebanese University, Beirut, Lebanon.
8 Faculty of Public Health, Section 1, Lebanese University, Beirut P.O. Box 6573, Lebanon
9 University of Balamand, Faculty of Arts and Sciences, Tripoli P.o. Box 100, Lebanon, tony.tannous@balamand.edu.lb
10 Department of Nutrition, Faculty of Pharmacy, Saint Joseph University of Beirut, Lebanon, joseph.matta@usj.edu.lb
11 Industrial Research Institute, Lebanese University Campus, Hadat Baabda, Lebanon
Comment 2:31 Avoid starting sentences with numbers.
Response 2: Thank you for your comment. The number "11" at the beginning of line 29 is part of the affiliation numbering and not the beginning of a sentence. We have reviewed the formatting to ensure this distinction is clear in the revised version.
Comment 3: 42 Delete green marks.
Response 3: Thank you for pointing out. The green marks were deleted from line 42
Comment 4: 93 Delete “According to”. “Replaces” instead of “changes”.
Response 4: Thank you for the suggestion. The phrase "According to" has been deleted and replaced with "As per", and "changes" has been corrected to "replaces" as recommended. The revised sentence, now found on lines 140–141, reads:
"As per the Commission Regulation (EU) 2023/915, which replaces Regulation (EC) No 1881/2006, raw milk and dairy products can contain up to 0.020 mg/kg and 0.005 mg/kg of lead and Cd respectively."
All changes have been highlighted in red in the revised manuscript.
Comment 5: 95 “Cadmium” shouldn’t be capitalized.
Response 5: Thank you for your observation. You are absolutely right “cadmium” should not be capitalized. In accordance with another reviewer’s suggestion, all instances of “cadmium” in the manuscript have been replaced with the abbreviation “Cd,” and the spelling has been standardized throughout the text and the changes have been highlighted in red in the revised manuscript
Comment 6:97 Should stay “Er et al.”.
Response 6: Thank you for your remark. The citation “Er C et” has been corrected to “Er et al.” and is now reflected accordingly in line 145 of the revised manuscript and highlighted in red.
Comment 7:146, 240, 254, 256, 259 Delete yellow marks.
Response 7: Thank you for your observation. The yellow marks in the mentioned lines have been removed in the revised version of the manuscript
Comment 8:159 Should be italic.
Response 8: Thank you for the remark. “Data analysis” has been italicized as recommended and is now found on line 347 of the revised manuscript and highlighted in Red.
Comment 9:183 Word “excellent” to be replaced with something more appropriate and in line with the text that is following it.
Response 9: Thank you for the suggestion. The word “excellent” has been replaced with “prime”, as this term aligns better with the following text of the manuscript and reflects the original wording used in Darwish et al.'s article. This correction can now be found in red in line 378.
Comment 10: 236 Should stay “There is little literature that has assessed lead and cadmium levels in yogurts.”.
Response 10: Thank you for your remark. The sentence “Little literature assessed the lead and cadmium levels in yogurts” has been revised to “There is little literature that has assessed lead and cadmium levels in yogurts,” as suggested. This correction is now found in line 451 and has been highlighted in red.
Comment 11: Table 3. Capital letters only for first words in sentences.
Response 11: Thank you for your comment. Only first words in sentences were capitalized
Comment 12: 323 Should not be bolded.
Response 12: Thank you for the observation. The formatting has been corrected - the bold font was removed, and the text now appears in regular style. This adjustment can be found in line 670 of the revised manuscript.
Comment 13: 324-450 Some references from the authors should be added to prove previous experience, if possible.
Response 13:Thank you for your comments. Some references from the authors were added indeed.
Round 2
Reviewer 1 Report
Comments and Suggestions for Authors
The manuscript has been significantly improved.
My comments concern the formatting of the manuscript, e.g. different font format and size, table 1 has no lines and should have horizontal lines, tables 3 and 7 have vertical and horizontal lines, missing periods in sentences (line 545). I suggest carefully analyzing the instructions for authors in Toxics and formatting the manuscript.
References
A lot of formatting errors.
My comment was misunderstood
e.g.
Is:
Exposure to heavy metals is one of the biggest risks to the biological system and, in particular, human health (Muneam et al.) [1]
Should be:
Exposure to heavy metals is one of the biggest risks to the biological system and, in particular, human health [1]
Is:
According to Hoteit et al., the average body weight for Lebanese adult consumers is 73.8 kg [46], therefore the daily intake.
Should be:
According to Hoteit et al., [46] the average body weight for Lebanese adult consumers is 73.8 kg therefore the daily intake.
Is:
Different studies reported the quantification of heavy metals, especially Pb and Cd, in milk and milk products worldwide. As an example, Er et al. evaluated the Pb levels in milk and yogurt samples in Turkey. They found that Pb levels were in the range of 15-61 ng/mL and 21-42 ng/g for milk and yogurt, respectively [27]. Besides, Winiarska-Mieczan et al. determined the Pb and Cd in yogurts and cream cheese samples in Poland [22]. In addition, Zhou et al. found that the levels of Pb and Cd were higher than the acceptable levels in milk samples, and this was attributed to industrial activities that probably led to milk contamination in China (Zhou et al.) [28]. Also, Yi-Jie Dai et al. reported the Pb and Cd contamination of Mish and Karish cheese samples, respectively, in Egypt [29]. Rebellato et al. determined Pb and Cd levels in yogurt samples in Brazil, and they were 21.58 and 4.20 μg/kg, respectively [30].
Should be:
Different studies reported the quantification of heavy metals, especially Pb and Cd, in milk and milk products worldwide. As an example, Er et al. [27] evaluated the Pb levels in milk and yogurt samples in Turkey. They found that Pb levels were in the range of 15-61 ng/mL and 21-42 ng/g for milk and yogurt, respectively [27]. Besides, Winiarska-Mieczan et al. [22] determined the Pb and Cd in yogurts and cream cheese samples in Poland. In addition, Zhou et al. [28] found that the levels of Pb and Cd were higher than the acceptable levels in milk samples, and this was attributed to industrial activities that probably led to milk contamination in China (Zhou et al.). Also, Yi-Jie Dai et al. [29] reported the Pb and Cd contamination of Mish and Karish cheese samples, respectively, in Egypt. Rebellato et al. [30] determined Pb and Cd levels in yogurt samples in Brazil, and they were 21.58 and 4.20 μg/kg, respectively.

Author Response
Comment 1: 1 has no lines and should have horizontal lines, tables 3 and 7 have vertical and horizontal lines, missing periods in sentences (line 545). I suggest carefully analyzing the instructions for authors in Toxics and formatting the manuscript.
Comment 2: Thank you for your valuable feedback. The formatting of all tables has been revised in accordance with the Toxics journal's guidelines. Specifically:
- Table 1 has been reformatted to include horizontal lines only (page 5, line 198)
- Tables 3 and 7 have been updated by removing all vertical lines, retaining only the required horizontal lines (page 15, line 475 and page 18, line 578 respectively)
- Missing periods and other minor punctuation issues, including the one on line 545 (which can still be found on line 545), have been corrected.
The manuscript has been carefully reviewed to ensure full alignment with the journal's formatting instructions.
Comment 2: References
A lot of formatting errors.
My comment was misunderstood
e.g.
Is:
Exposure to heavy metals is one of the biggest risks to the biological system and, in particular, human health (Muneam et al.) [1]
Should be:
Exposure to heavy metals is one of the biggest risks to the biological system and, in particular, human health [1]
Is:
According to Hoteit et al., the average body weight for Lebanese adult consumers is 73.8 kg [46], therefore the daily intake.
Should be:
According to Hoteit et al., [46] the average body weight for Lebanese adult consumers is 73.8 kg therefore the daily intake.
Is:
Different studies reported the quantification of heavy metals, especially Pb and Cd, in milk and milk products worldwide. As an example, Er et al. evaluated the Pb levels in milk and yogurt samples in Turkey. They found that Pb levels were in the range of 15-61 ng/mL and 21-42 ng/g for milk and yogurt, respectively [27]. Besides, Winiarska-Mieczan et al. determined the Pb and Cd in yogurts and cream cheese samples in Poland [22]. In addition, Zhou et al. found that the levels of Pb and Cd were higher than the acceptable levels in milk samples, and this was attributed to industrial activities that probably led to milk contamination in China (Zhou et al.) [28]. Also, Yi-Jie Dai et al. reported the Pb and Cd contamination of Mish and Karish cheese samples, respectively, in Egypt [29]. Rebellato et al. determined Pb and Cd levels in yogurt samples in Brazil, and they were 21.58 and 4.20 μg/kg, respectively [30].
Should be:
Different studies reported the quantification of heavy metals, especially Pb and Cd, in milk and milk products worldwide. As an example, Er et al. [27] evaluated the Pb levels in milk and yogurt samples in Turkey. They found that Pb levels were in the range of 15-61 ng/mL and 21-42 ng/g for milk and yogurt, respectively [27]. Besides, Winiarska-Mieczan et al. [22] determined the Pb and Cd in yogurts and cream cheese samples in Poland. In addition, Zhou et al. [28] found that the levels of Pb and Cd were higher than the acceptable levels in milk samples, and this was attributed to industrial activities that probably led to milk contamination in China (Zhou et al.). Also, Yi-Jie Dai et al. [29] reported the Pb and Cd contamination of Mish and Karish cheese samples, respectively, in Egypt. Rebellato et al. [30] determined Pb and Cd levels in yogurt samples in Brazil, and they were 21.58 and 4.20 μg/kg, respectively.
Response 2: Thank you for your observation, and apologies for the earlier misunderstanding of your comment. We now fully understand the intended formatting standard for in-text citations, and all such instances have been revised to align with the required style. Specifically:
- Exposure to heavy metals is one of the biggest risks to the biological system and, in particular, human health (Muneam et al.) [1]
was corrected to:
Exposure to heavy metals is one of the biggest risks to the biological system and, in particular, human health [1] (page 2, lines 59–60). - According to Hoteit et al., the average body weight for Lebanese adult consumers is 73.8 kg [46], therefore the daily intake.
Was corrected to:
According to Hoteit et al., [46] the average body weight for Lebanese adult consumers is 73.8 kg therefore the daily intake. (page 17, line 545)
- Different studies reported the quantification of heavy metals, especially Pb and Cd, in milk and milk products worldwide. As an example, Er et al. evaluated the Pb levels in milk and yogurt samples in Turkey. They found that Pb levels were in the range of 15-61 ng/mL and 21-42 ng/g for milk and yogurt, respectively [27]. Besides, Winiarska-Mieczan et al. determined the Pb and Cd in yogurts and cream cheese samples in Poland [22]. In addition, Zhou et al. found that the levels of Pb and Cd were higher than the acceptable levels in milk samples, and this was attributed to industrial activities that probably led to milk contamination in China (Zhou et al.) [28]. Also, Yi-Jie Dai et al. reported the Pb and Cd contamination of Mish and Karish cheese samples, respectively, in Egypt [29]. Rebellato et al. determined Pb and Cd levels in yogurt samples in Brazil, and they were 21.58 and 4.20 μg/kg, respectively [30].
Was corrected to:
Different studies reported the quantification of heavy metals, especially Pb and Cd, in milk and milk products worldwide. As an example, Er et al. [27] evaluated the Pb levels in milk and yogurt samples in Turkey. They found that Pb levels were in the range of 15-61 ng/mL and 21-42 ng/g for milk and yogurt, respectively [27]. Besides, Winiarska-Mieczan et al. [22] determined the Pb and Cd in yogurts and cream cheese samples in Poland. In addition, Zhou et al. [28] found that the levels of Pb and Cd were higher than the acceptable levels in milk samples, and this was attributed to industrial activities that probably led to milk contamination in China (Zhou et al.). Also, Yi-Jie Dai et al. [29] reported the Pb and Cd contamination of Mish and Karish cheese samples, respectively, in Egypt. Rebellato et al. [30] determined Pb and Cd levels in yogurt samples in Brazil, and they were 21.58 and 4.20 μg/kg, respectively (page 4 , line 139-151)
Reviewer 2 Report
Comments and Suggestions for Authors
Dear Editor,
The authors have improved the manuscript. The inclusion of EDI has brought a more accurate risk assessment. The statistical analyses are now solid. I suggest only a small edit in Figure 1.
In my sixth comment, I mention the geographical location of Lebanon. I was referring to Figure 1. I think it would be interesting to upgrade the map by including the country's location in relation to the continent/world. In any case, I congratulate the authors on their efforts to improve the manuscript.
Author Response
Comment 1: Dear Editor,
The authors have improved the manuscript. The inclusion of EDI has brought a more accurate risk assessment. The statistical analyses are now solid. I suggest only a small edit in Figure 1.
In my sixth comment, I mention the geographical location of Lebanon. I was referring to Figure 1. I think it would be interesting to upgrade the map by including the country's location in relation to the continent/world. In any case, I congratulate the authors on their efforts to improve the manuscript.
Response 1: We sincerely thank the reviewer for the valuable feedback and positive evaluation of the revised manuscript. In response to the sixth comment, we have updated Figure 1 to include Lebanon’s location in relation to the continent and the world, thereby providing clearer geographic context for readers unfamiliar with the region. We appreciate your thoughtful suggestion and your kind words regarding the improvements made to the manuscript.
The revised Figure 1 can be found in page 6, line 200 and the revised title of the figure is : View of Lebanon’s location on the global map with the geographic distribution of the yogurt sampling sites across the eleven Lebanese Regions.